# Dynamics of Onset and Progression in Amyotrophic Lateral Sclerosis

**DOI:** 10.3390/brainsci15060601

**Published:** 2025-06-03

**Authors:** Michael Swash, Mamede de Carvalho

**Affiliations:** 1Barts and the London School of Medicine, Queen Mary University of London, London E1 4NS, UK; mswash@btinternet.com; 2Department of Neurosciences and Mental Health, Centro Hospitalar Universitário Lisboa Norte, 1649-035 Lisboa, Portugal; 3Faculdade de Medicina, Instituto de Fisiologia, Universidade de Lisboa, 1649-028 Lisbon, Portugal

**Keywords:** amyotrophic lateral sclerosis, lower motor neuron, onset, progression, upper motor neuron, treatment

## Abstract

This review focuses on the complexities of amyotrophic lateral sclerosis (ALS) onset, highlighting the insidious nature of the disease and the challenges in defining its precise origin and early pathogenic mechanisms. The clinical presentation of ALS is characterised by progressive muscle weakness and wasting, often with widespread fasciculations, reflecting lower motor neuron hyperexcitability. The disease’s pathogenesis involves a prolonged preclinical phase of neuronal proteinopathy, particularly TDP-43 accumulation, which eventually leads to motor neuron death and overt ALS. This review discusses the difficulties in detecting this transition and the implications for early therapeutic intervention. It also addresses the involvement of both the upper and lower motor neuron systems, as well as the importance of following presymptomatic patients with genetic mutations. The significance of understanding the distinct processes of TDP-43 deposition and subsequent neuronal degeneration in developing effective treatments is emphasised.

## 1. Introduction

Amyotrophic lateral sclerosis (ALS) is a rare disease (prevalence is about 5–6/100,000) that begins with progressive weakness and wasting in a relatively focal distribution, sometimes even resembling a root lesion [1]. Its spread to other regions generally follows a pattern of regional proximity [2,3]. Widespread fasciculation, often an early and persistent feature, indicates hyperexcitability of lower motor neurons (LMN) in other, already affected motor regions [4,5]. The onset is insidious, e.g., with new-onset muscular cramps, fatigue, or reduced exercise performance, often recognised only in retrospect [6]. This evolving clinical pattern has raised questions regarding the development of the illness and specifically the notion of a supposed discrete site of onset, currently centred on the hypothesis of cortical hyperexcitability as a primary abnormality. There are fundamental related questions concerning the pathobiology of the disease, which must underlie any physiological change, especially the duration of a preclinical phase of neuronal proteinopathy tipping into cytological toxicity, inducing functional disturbance and motor-neuronal death, i.e., diagnosable ALS [7]. This transition between preclinical proteinopathy and progressive cell death is currently undefinable [8]. It is an important concept since it constitutes the earliest potential opportunity to introduce a therapy, if such were available. The recognition of a disease-associated genetic mutation itself raises the possibility of preclinical disease-modulating management before the phase of progressive neuronal death commences [9]. A discussion of the site of origin of ALS [10,11]—whether affecting the motor cortex, the spinal anterior horns, or, more likely, both, as well as other motorneuronal control systems—may therefore not be fundamentally important. Imaging, physiological, and pathological studies show the involvement of the upper motor neuron (UMN) systems in the frontal and prefrontal cortex, corpus callosum, and deep brain white matter [9] and of LMN neurons, interneurons, and small gamma motor neurons in the spinal anterior horns [12], while the sensory systems are spared. The frequently associated frontal lobe syndrome is best regarded as the degeneration of a related effector brain system. Clinical experience suggests that LMN degeneration usually determines the clinical presentation [13].

The progressive phase of sporadic ALS, when clinical diagnosis becomes possible, is considered to be due to metabolic stresses associated with the prior gradual deposition of aggregated TDP-43 protein in the neuronal cytosol, mislocalised from its usual location in the nucleus. This is an almost universal feature of sporadic ALS [14]. Unique proteinopathies are also a feature of SOD1- and FUS-related familial ALS [13].

**Clinical Vignette**—A 51-year-old woman presented with intermittent, bilateral hand tingling of a 4-month duration, vague gait imbalance, and a fall a few weeks earlier. There was a history of recovered Bell’s palsy, well-controlled epilepsy since childhood, and mild hypertension. Her family history was unremarkable. Examination revealed mild right peripheral facial paresis. There was normal muscle strength, bulk, and tone and a normal sensory examination in all regions. Speech and swallowing were normal. There was no ataxia. Tendon reflexes were slightly brisk in all four limbs, but the plantar responses were flexor and the Hoffman sign was absent. Routine blood tests were within normal limits. Upper and lower limb motor and sensory nerve conduction studies, and needle electromyography (EMG) of the upper and lower limb muscles, were normal (MdC). Brain and spinal cord magnetic resonance imaging (MRI) was unremarkable.

Two years later, progressive left foot drop, without sensory symptoms, developed quite rapidly, soon spreading to the right leg. Spinal MR imaging was again unremarkable. Marked weakness and atrophy developed in both legs and less markedly in the upper limbs. The tendon reflexes were increased, with bilateral Hoffman signs and a right extensor plantar response. Fasciculations were seen in the tongue, but not in the limb muscles. Sensory examination and cognition were normal. She required a cane to walk. Repeat motor and sensory nerve conduction studies were normal in the upper and lower limbs, and needle EMG (MdC) showed fasciculation potentials with chronic neurogenic change in the upper and lower limb muscles. Four months later, she was dysarthric and wheelchair-confined, with marked functional impairment in both upper limbs. There were no sphincteric symptoms.

## 2. Learning from Clinical Observations

This patient presented with a minor neurological disorder that defied diagnosis, which was not, at that time, suggestive of motor neuron disease, but a diagnosis of ALS became evident 2 years later. The opportunity to assess a patient at a relatively short time before the onset of overt ALS is infrequent, and there is little information about subtle symptoms and signs during this period of 1–5 years before diagnosis [6,15]. In a prospective study of subjects with G93A SOD1 mutations, initially unaffected by ALS, rapid reductions in the motor unit number (MUNE) in the extensor digitorum communis and abductor pollicis brevis muscles occurred 16 months and 9 months, respectively, before the onset of weakness [16]. This is consistent with Wohlfart’s conclusion that compensatory reinnervation fails only when at least 30% of the motor neurons in an anterior horn pool have been lost [17]. We previously described two other patients presenting with persistent back and neck pain, when EMGs were normal. Lower limb weakness developed nearly 2 years later, when EMGs showed widespread chronic partial denervation with fasciculations. Both patients progressed rapidly, with death 2 years later [18]. In addition, fasciculations often precede neurogenic change on EMG [19,20]. These few case reports demonstrate ALS onset but do not define it [6]. Clinical observations cannot provide markers to address the presumed primary preclinical phase of cytosolic proteinopathy in the motor systems of the brain.

Although the onset of ALS is clinically asymmetric and even focal, EMG abnormalities are typically more widespread [18,20,21]. In the course of the disease, spontaneous phases of relatively slower or faster progression may occur [22], and surgical procedures may induce more rapid progression [23]. The disease appears to spread through the central nervous system (CNS), both by regional proximity and through UMN pathways [3]. The localised progression of ALS within cortical and spinal regions can manifest in distinct patterns: somatotopic spreading within the cortex (implying spread to adjacent body representations) and contralateral spreading in spinal regions [3]. A significant clinical observation is that UMN involvement is linked to the faster vertical spread of the disease in ALS patients [3].

In SOD1 ALS, cortical hyperexcitability has been reported 3–8 months before clinical onset [24], matching the preclinical reduction in MUNE. Increased serum and CSF neurofilament levels have been reported up to two years before clinical onset in *C9orf72* carriers, consistent with early axonal degeneration [15,25] but not predicting the progression rate. Furthermore, these observations do not imply any specific sequence of motor pathway degeneration, and other authors have not confirmed a consistent presymptomatic neurofilament increase [26]. The duration of a possible period of neuronal dysfunction that might be *reversible*, as distinct from motor neuron death in ALS, is unknown. Nonetheless, this is clearly an important factor in any proposed therapeutic strategy. MRI studies have revealed white matter (frontal and orbitofrontal regions, thalamic radiation, corpus callosum, corticospinal tract) and grey matter changes (motor cortex, thalamus, cerebellum) in presymptomatic *C9orf72* carriers [9], perhaps consistent with the proposed long preclinical phase [7]. However, these changes were not consistent in all studies [9], indicating pathobiological variability in the disease. In a Swedish study of >1.8 million healthy military recruits, low physical strength and low haematocrit at the conscription physical examination were proposed as risk factors for the development of ALS several decades later [27]. Swash and Ingram [6] described preclinical and reversible motor features in ALS nearly 40 years ago, but these observations have not been integrated into the understanding of the onset and progression of the ALS syndrome. Thus, the preclinical phase of ALS is not pathologically quiescent and may be very long. Preclinical studies in sporadic ALS are obviously difficult to accomplish since case ascertainment is currently impossible. However, the lifetime risk in first-degree relatives of Irish individuals with ALS is increased about four times (1.4% vs. 0.3%), perhaps suggesting a possible novel approach to this especially difficult problem [28].

The *C9orf72* hexanucleotide repeat expansion is the most common genetic cause of both ALS and frontotemporal dementia (FTD), establishing a significant overlap between these neurodegenerative disorders [29]. Patients with this expansion can present with pure ALS, pure FTD (most commonly behavioural variant FTD), or a combination of both (ALS-FTD), even within the same family, highlighting a disease spectrum [29]. This condition follows an autosomal dominant inheritance pattern with incomplete penetrance, estimated to be around 20–50%, although this varies between families [29]. The reasons for this significant phenotypic and penetrance variability are still unclear. However, potential factors include somatic expansions occurring only in the brain [30], the influence of multiple genes (oligogenicity), and environmental factors [29]. Unravelling this complexity will require further investigation, including established “wet” biomarkers like neurofilaments and dipeptide repeat proteins, as well as newer ones such as ubiquitin carboxyl-terminal hydrolase isozyme L1 [31]. Brain imaging and neurophysiological markers will also be crucial. For instance, as mentioned above, asymptomatic carriers of the *C9orf72* expansion can show structural and functional changes on MRI years before symptoms emerge [9,32], and patients with this mutation often display signs of abnormal cortical hyperexcitability in the motor system when assessed with transcranial magnetic stimulation [33].

It is intriguing to consider that some motor neuron diseases, such as Hirayama disease, exhibit limited temporal and topographic clinical expression [34]. Similarly, some patients with widespread fasciculations and motor unit loss never progress to typical ALS [35,36]. These observations suggest that disease progression is governed by an unfavourable biological balance.

## 3. Clinical Implications

The undefined preclinical phase and the subsequent clinical onset and progression of ALS may represent different biological processes, based on a common theme of cytoplasmic proteinaceous inclusion material in motor neurons and possibly also in glial cells (Figure 1) [7,14,37,38,39]. The ubiquitinated inclusions of pathological TDP-43 accumulation in motor neurons found in sporadic ALS and in *C9orf72* G_4_C_2_-repeat ALS, as well as other specific proteinopathies, as in SOD1- and FUS-related ALS, suggest that cytosolic proteinopathy is the initial pathological marker, developing long before the transition to overt ALS [14]. There is evidence of widespread motor system-related brain involvement, and generally the more substantial involvement of the final common pathway of LMNs than UMNs [14], once the disease is established and progressive, but the pattern of motor system degeneration varies substantially from case to case, as first recognised many years ago [40,41]. However, we suggest that mathematical models of spreading patterns for specific ALS phenotypes should be explored as outcomes in future clinical trials [3].

The direct correlation of this pathological model with corticomotor neurophysiological evidence is not currently possible, since the comparison of MRI, EMG, and cortical excitability studies should be considered critically due to the uncertain comparative sensitivity and relevance of these investigative techniques. When the degenerative process associated with neuronal proteinopathy ‘tips into’ overt progressive ALS, loss of function in UMN and LMN systems within the CNS develops rapidly, often seemingly increasingly rapidly, until denervation is very severe [4]. Thus, following the clinical detection of early LMN dysfunction, there is only a short time frame available for intervention with any potentially effective neuroprotective therapy. Therefore, in both sporadic and *C9orf72* mutation-associated ALS, therapy directed toward TDP-43 neuronal dysmetabolism, causing neuronal cytoplasmic inclusions, is likely to be critically important [14]. Currently, clinical trials are directed to the management of the ill-understood cascade of cytosolic metabolic abnormality, associated with excitotoxicity, that leads to motor neuronal death and overt clinical ALS. Transgenic models and clinical studies suggest that microRNAs and antisense nucleotides may offer a means of reducing the cytoplasmic load of abnormal proteins in genetically determined ALS [42,43], but this remains uncertain given the current inability to recognise the long preclinical phase of sporadic ALS, even in those with genetic predispositions, and whether this will prevent the catastrophic onset of neurodegeneration that characterises ALS itself.

The forthcoming era of adaptive platform clinical trials [44] necessitates the inclusion of sporadic patients presenting with very early disease manifestations and mutation carriers exhibiting minimal clinical signs. A rigorous, systematic evaluation incorporating biomarkers (e.g., “wet” biomarkers), advanced imaging, and neurophysiological assessments will be indispensable in determining the therapeutic efficacy. Furthermore, trial protocols will be meticulously optimised through the application of artificial intelligence models.

## 4. Detecting Those at Risk for ALS

The preclinical detection of people likely to develop ALS would be a major advance. This might possibly be facilitated by an improved understanding of genetic or even environmental risk factors, but whether there are any such significant factors is disappointingly uncertain. Subtle neurological symptoms, as in our clinical vignette, might be unexpectedly important. Minor associations have been reported with smoking, physical exercise, various dietary factors, and a high basal metabolic rate or low body mass index before diagnosis [45]. In addition, environmental industrial pollution curiously parallels the first descriptions of ALS and the increasing incidence of the disease since the mid-nineteenth century [46]. However, none of these represent predictable risk factors. A six-step process leading to sporadic ALS, without implications regarding a temporal gradient, has been suggested, calculated from the slope of a regression analysis of log incidence against log age at diagnosis [47,48]. A smaller number of steps was calculated as required for ALS associated with a *C9orf72* GGGGCC repeat expansion (three steps), *TARDBP* (four steps), and *SOD1* (two steps) mutations [49], suggesting different causative processes. These hypothesised ‘steps’ are not necessarily independent, discrete exposures or factors but represent cumulative pathogenic influences that are not sufficient to cause ALS if acting alone. In addition to fronto-temporal lobar neurodegeneration, four clinical classes of ALS cases have been recognised, i.e., patients with combined UMN and LMN features; those with predominant LMN disorder; those with associated fronto-hippocampal degeneration; and those with brainstem and basal ganglia involvement [14,50]. However, these are conventionally treated as homologous in clinical studies, including clinical trials, although these differences must be pathobiologically meaningful. They are consistent with ALS as a complex multi-system CNS disorder [51,52,53,54]. The traditional distinction between genetic and sporadic ALS and the differing clinical phenotypes of the disease is itself blurred [50]. We emphasise that the patient described in this report first presented with vague, apparently non-motor symptoms, suggesting that a detailed enquiry into such events months or years before the onset of overt ALS could be worthwhile [15,44,45].

## 5. A Way Forward

We suggest that the key to both the prevention and management of sporadic ALS will require an understanding of the separate processes of the motor-neuronal cytosolic deposition of TDP-43 and the several metabolic cellular derangements that lead to inexorable motor neuronal death, including their associated inhibitory interneurons, both in the UMN and LMN systems and in prefrontal neuronal systems, as well as in related neuronal systems in the basal ganglia and the spinocerebellar nuclei of Clarke’s column. Physiological evidence of hyperexcitability in the motor cortex is best considered a result of the pathologic features of the disease and not as direct causation. It has proven difficult to interrupt the process of neuronal death. This may require first clearing the cytosol of aberrant, misfolded, and ubiquitinated TDP-43—for example, by antisense nucleotide therapies [43]. ALS does not begin at the onset of motor or frontal symptoms but long before these events. The currently designed clinical trials do not address the proteinopathy and are probably applied too late in the natural history of the disease [55]. Moving forward, effective therapeutic strategies for ALS will likely depend on early intervention targeting the underlying TDP-43 proteinopathy and the complex metabolic dysregulations before widespread neurodegeneration occurs.

## Figures and Tables

**Figure 1 brainsci-15-00601-f001:**
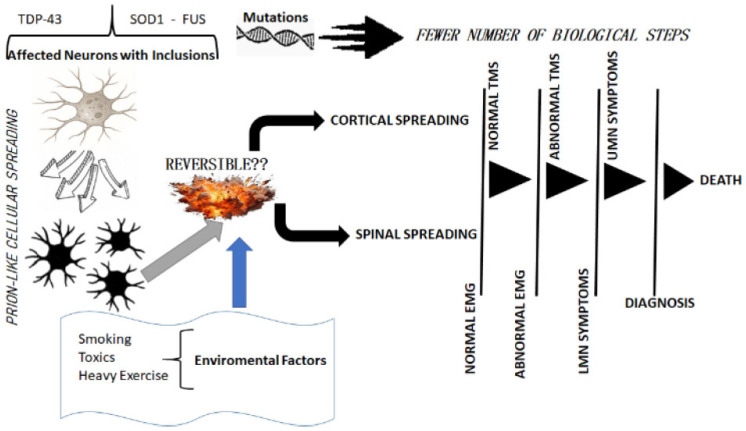
A schematic view of the main steps of disease progression. The figure shows the pathogenic pathway for ALS, from neuronal toxic aggregates to clinical and neurophysiological features, highlighting the role of genetic mutations (associated with fewer pathogenic steps) and environmental risk factors.

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
