# Peer review of "Dynamics of Onset and Progression in Amyotrophic Lateral Sclerosis"

_brainsci, 2025, doi:10.3390/brainsci15060601_

Round 1
Reviewer 1 Report
Comments and Suggestions for Authors
The authors are both accomplished ALS researchers and have put together a well organized succinct and important article extolling the need for pre-symptomatic disease recognition and a dual and earlier approach to therapeutic intervention to stave off, or even reverse, disease progression by addressing the cytosolic proteinopathy as well as upstream metabolic events. It is important to recognize as the authors state that the preclinical phase of ALS is not pathologically quiescent and could be very long, offering opportunities for interventions that may be more effective.
I have no major concerns with the article, though some small suggestions as follows.
There is a typo in line 42
I would specify LMN facial paresis in line 63
Line 99 should say, are “typically” more widespread, as what is stated is not always the case. Likewise in line 100, “frequently” should be changed to “may,” especially given the ALSFRS-R being the determining factor for judging improvement/worsening.
It would be helpful to have at least a line explaining what heritability is/isn’t given the high risk of the line about heritability being misconstrued (line 125), so that readers don’t think the relatives of Irish individuals with ALS have a 50% chance of getting ALS.
Stylistically, I might move the very last line deeper into the paragraph/paper, as it seems abrupt ending on that line.
Author Response
#R1
We are very grateful for the excellent comments from this reviewer that strongly contributed to improve que quality of our manuscript
The authors are both accomplished ALS researchers and have put together a well-organized succinct and important article extolling the need for pre-symptomatic disease recognition and a dual and earlier approach to therapeutic intervention to stave off, or even reverse, disease progression by addressing the cytosolic proteinopathy as well as upstream metabolic events. It is important to recognize as the authors state that the preclinical phase of ALS is not pathologically quiescent and could be very long, offering opportunities for interventions that may be more effective.
The authors are grateful for this reviewer positive comments
I have no major concerns with the article, though some small suggestions as follows.
There is a typo in line 42
The typos were checked and corrected
I would specify LMN facial paresis in line 63
The sentence was changed to “Examination revealed a mild right peripheral facial paresis”.
Line 99 should say, are “typically” more widespread, as what is stated is not always the case. Likewise in line 100, “frequently” should be changed to “may,” especially given the ALSFRS-R being the determining factor for judging improvement/worsening.
We are grateful for these correction, which were done.
It would be helpful to have at least a line explaining what heritability is/isn’t given the high risk of the line about heritability being misconstrued (line 125), so that readers don’t think the relatives of Irish individuals with ALS have a 50% chance of getting ALS.
Thanks, we have changed this paragraph “However, the lifetime risk in first-degree relatives of Irish individuals with ALS is increased about 4 times (1.4% vs 0.3%), perhaps suggesting a possible novel approach to this especially difficult problem.”
Stylistically, I might move the very last line deeper into the paragraph/paper, as it seems abrupt ending on that line.
We agree, the final section was changed “We suggest that the key to both prevention and management of sporadic ALS will require understanding of the separate processes of motor neuronal cytosolic deposition of TDP-43, and the several metabolic cellular derangements that lead to inexorable motor neuronal death, including their associated inhibitory interneurons, both in the UMN and LMN systems and in prefrontal neuronal systems, as well as in related neuronal systems in basal ganglia and the spinocerebellar nuclei of Clarke’s column. Physiological evidence of hyperexcitability in motor cortex is best considered a result of the pathologic features of the disease, not a direct causation. It has proven difficult to interrupt the process of neuronal death. This may require first clearing the cytosol of aberrant, misfolded and ubiquitinated TDP-43, for example by antisense nucleotide therapies.33 ALS does not begin at onset of motor or frontal symptoms, but long before those events. Currently designed clinical trials do not address the proteinopathy and are probably applied too late in the natural history of the disease.44 Moving forward, effective therapeutic strategies for ALS will likely depend on early intervention targeting the underlying TDP-43 proteinopathy and the complex metabolic dysregulations before widespread neurodegeneration occurs.”
Reviewer 2 Report
Comments and Suggestions for Authors
The paper by Michael Swash and Mamede de Carvalho is a brief narrative review that explores the onset and progression dynamics of ALS, providing insights of great interest for future research that will be essential, as is already the case for other neurodegenerative diseases (e.g. Alzheimer's disease), in identifying patients at stages when therapies could be effective. The work is engaging and thought-provoking. I have a few suggestions for the Authors to cover some aspects that are not currently explored in depth:
- I suggest further exploration of the dynamics that differentiate the modes of progression depending on the site of onset. Indeed, this may not be so decisive. Still, several studies have shown that when neuropathological damage begins at the cortical level, it spreads to adjacent somatotopic representations (e.g., leading to hemiparesis), while when it starts from the second motor neuron, the tendency is to pass directly to the contralateral side. What could be the implications in terms of the early identification of patients? What aspects are related to the vulnerability of the affected neurons and, therefore, potential therapeutic targets?
- The ALS-FTD continuum also deserves further investigation. Why, with the same mutation (e.g., C9orf72), do some patients have overt motor neuron disease and minimal cognitive symptoms, while others have only minimal motor changes and prevalent cognitive and behavioural disorders? Recent work has explored FTD and suggested how to deal with its diagnostic complexity, including analyses of biomarkers (e.g. NIBS, autonomic neurophysiology, blood biomarkers) that could be very useful for exploring differences in clinical trajectories and identifying patients in early stages, especially considering the significance of cortical hyperexcitability (in particular NIBS or other neurophysiological techniques). I suggest citing these works and also recent literature reviews as further reading for the reader;
- There is a reference to Figure 1, which is not included. A figure or table would be useful to show the early alterations that should be investigated to identify patients early, and other key concepts discussed in this review;
- One aspect that I believe is worth exploring further is cases of motor neuron disease in which neuropathological damage does not progress, e.g., 10.3389/fneur.2020.00183. I suggest citing this work and others like it and evaluating the differences between these patients and those typical of the disease described in the review. What does this disparity in behaviour suggest?
- Another aspect that I believe could be very useful is further exploration of environmental risk factors, which could be key to identifying patients in the early stages. For example, studies have been conducted on Amyotrophic Lateral Sclerosis and air pollutants with ecological studies. Other studies have explored other known factors, such as pesticides, occupational activities, etc. Presenting these aspects could also be very useful in suggesting how to identify patients who are potentially at higher risk in addition to those with genetic factors.
Author Response
#R2
We are very grateful for the excellent comments from this reviewer that strongly contributed to improve que quality of our manuscript
The paper by Michael Swash and Mamede de Carvalho is a brief narrative review that explores the onset and progression dynamics of ALS, providing insights of great interest for future research that will be essential, as is already the case for other neurodegenerative diseases (e.g. Alzheimer's disease), in identifying patients at stages when therapies could be effective. The work is engaging and thought-provoking. I have a few suggestions for the Authors to cover some aspects that are not currently explored in depth:~
The authors are grateful for this reviewer positive comments
- I suggest further exploration of the dynamics that differentiate the modes of progression depending on the site of onset. Indeed, this may not be so decisive. Still, several studies have shown that when neuropathological damage begins at the cortical level, it spreads to adjacent somatotopic representations (e.g., leading to hemiparesis), while when it starts from the second motor neuron, the tendency is to pass directly to the contralateral side. What could be the implications in terms of the early identification of patients? What aspects are related to the vulnerability of the affected neurons and, therefore, potential therapeutic targets?
We are grateful, in “Learning from clinical observations” we added “The localized progression of ALS within cortical and spinal regions can manifest in distinct patterns: somatotopic spreading within the cortex (implying spread to adjacent body representations) and contralateral spreading in spinal regions. A significant clinical observation is that UMN involvement is linked to a faster vertical spread of the disease in ALS patients..”
- The ALS-FTD continuum also deserves further investigation. Why, with the same mutation (e.g., C9orf72), do some patients have overt motor neuron disease and minimal cognitive symptoms, while others have only minimal motor changes and prevalent cognitive and behavioural disorders? Recent work has explored FTD and suggested how to deal with its diagnostic complexity, including analyses of biomarkers (e.g. NIBS, autonomic neurophysiology, blood biomarkers) that could be very useful for exploring differences in clinical trajectories and identifying patients in early stages, especially considering the significance of cortical hyperexcitability (in particular NIBS or other neurophysiological techniques). I suggest citing these works and also recent literature reviews as further reading for the reader;
In “Learning from clinical observations” we added “The C9orf72 hexanucleotide repeat expansion is the most common genetic cause of both ALS and frontotemporal dementia (FTD), establishing a significant overlap between these neurodegenerative disorders. Patients with this expansion can present with pure ALS, pure FTD (most commonly behavioural variant FTD), or a combination of both (ALS-FTD), even within the same family, highlighting a disease spectrum. This condition follows an autosomal dominant inheritance pattern with incomplete penetrance, estimated to be around 20–50%, though this varies between families. The reasons for this significant phenotypic and penetrance variability are still unclear. However, potential factors include somatic expansions occurring only in the brain, the influence of multiple genes (oligogenicity), and environmental factors. Unravelling this complexity will require further investigation including established "wet" biomarkers like neurofilaments and dipeptide repeat proteins, as well as newer ones such as ubiquitin carboxyl-terminal hydrolase isozyme L1. Brain imaging and neurophysiological markers will also be crucial. For instance, asymptomatic carriers of the C9orf72 expansion can show structural and functional changes on MRI years before symptoms emerge, and patients with this mutation often display signs of abnormal cortical hyperexcitability in the motor system when assessed with transcranial magnetic stimulation,” including recent references.
- There is a reference to Figure 1, which is not included. A figure or table would be useful to show the early alterations that should be investigated to identify patients early, and other key concepts discussed in this review;
A figure was added to clarify the pathogenesis
- One aspect that I believe is worth exploring further is cases of motor neuron disease in which neuropathological damage does not progress, e.g., 10.3389/fneur.2020.00183. I suggest citing this work and others like it and evaluating the differences between these patients and those typical of the disease described in the review. What does this disparity in behaviour suggest?
In “Learning from clinical observations” we added in the end “It is intriguing to consider that some motor neuron diseases, such as Hirayama disease, exhibit a limited temporal and topographic clinical expression. Similarly, some patients with widespread fasciculations and motor unit loss never progress to typical ALS. These observations suggest that disease progression is governed by an unfavorable biological balance.”
- Another aspect that I believe could be very useful is further exploration of environmental risk factors, which could be key to identifying patients in the early stages. For example, studies have been conducted on Amyotrophic Lateral Sclerosis and air pollutants with ecological studies. Other studies have explored other known factors, such as pesticides, occupational activities, etc. Presenting these aspects could also be very useful in suggesting how to identify patients who are potentially at higher risk in addition to those with genetic factors.
These aspects were present in the previous version in “Detecting those at risk for ALS”. We decided not to expand this point further.
Reviewer 3 Report
Comments and Suggestions for Authors
The authors should consider the following points to improve the quality of submitted review
1- many abbreviations are not fully identified
2- authors should include prevelance of ALS
3-bizzare to include a case report within the review
4- the review misses illustrating figures to clarify the pathophysiology
5-recommendations and consideration for suggested treatment protocol should gain more attention
Author Response
#R3
We are very grateful for the excellent comments from this reviewer that strongly contributed to improve que quality of our manuscript
The authors should consider the following points to improve the quality of submitted review
- many abbreviations are not fully identified
This was corrected
- authors should include prevalence of ALS
This information was added in Introduction
3-bizzare to include a case report within the review
We've kept the case report in the text because it clearly illustrates the discussed points and was well-received by the other reviewers.
- the review misses illustrating figures to clarify the pathophysiology
A figure was added to clarify the pathogenesis
5-recommendations and consideration for suggested treatment protocol should gain more attention
In the end of “Clinical Implications” the following was added. “The forthcoming era of adaptive platform clinical trials necessitates the inclusion of sporadic patients presenting with very early disease manifestations and mutation carriers exhibiting minimal clinical signs. A rigorous, systematic evaluation incorporating biomarkers (e.g., "wet" biomarkers), advanced imaging, and neurophysiological assessments will be indispensable for detecting therapeutic efficacy. Furthermore, trial protocols will be meticulously optimized through the application of artificial intelligence models.”
Round 2
Reviewer 2 Report
Comments and Suggestions for Authors
No further comments from my side, thus the manuscript could be accepted in its current form.